# Effects of Cellulose Nanocrystals and Cellulose Nanofibers on the Structure and Properties of Polyhydroxybutyrate Nanocomposites

**DOI:** 10.3390/polym11122063

**Published:** 2019-12-11

**Authors:** Bobo Zhang, Chongxing Huang, Hui Zhao, Jian Wang, Cheng Yin, Lingyun Zhang, Yuan Zhao

**Affiliations:** College of Light Industry & Food Engineering, Guangxi University, Nanning 530004, China; Zbobo05@163.com (B.Z.); wjcj125521@163.com (J.W.); yc666yc@163.com (C.Y.); zy199113@163.com (Y.Z.)

**Keywords:** polyhydroxybutyrate (PHB), cellulose nanocrystals (CNCs), cellulose nanofibers (CNFs), mechanical properties

## Abstract

One of the major obstacles for polyhydroxybutyrate (PHB), a biodegradable and biocompatible polymer, in commercial applications is its poor elongation at break (~3%). In this study, the effects of nanocellulose contents and their types, including cellulose nanocrystals (CNCs) and cellulose nanofibers (CNFs) on the crystallization, thermal, and mechanical properties of PHB composites were systematically compared. We explored the toughening mechanisms of PHB by adding CNCs and cellulose CNFs. The results showed that when the morphology of bagasse nanocellulose was rod-like and its content was 1 wt %, the toughening modification of PHB was the best. Compared with pure PHB, the elongation at break and Young’s modulus increased by 91.2% and 18.4%, respectively. Cellulose nanocrystals worked as heterogeneous nucleating agents in PHB and hence reduced its crystallinity and consequently improved the toughness of PHB. This simple approach could potentially be explored as a strategy to extend the possible applications of this biopolymer in packaging fields.

## 1. Introduction

In the past few decades, the negative environmental impacts caused by petroleum-based polymers and the depletion of petroleum resources have been an important issue. Therefore, the research and development of materials based on renewable resources, such as biodegradable polymers, is imperative [1,2].

Polyhydroxybutyrate (PHB)—part of the polyhydroxyalkanoate (PHA) aliphatic polyester class—is a semi-crystalline polymer produced through numerous mechanisms, including the bacterial fermentation of sugars [3]. PHB is a water-insoluble biodegradable polymer with high crystallinity, biocompatibility, biodegradability, permeability, and processability [4]. PHB is potentially useful for biodegradable materials, such as mulch film, absorbable surgical sutures for drug delivery, and tissue engineering due to its biodegradability and biocompatibility [5]. Nevertheless, these potential applications of PHB have been limited due to its severe brittleness, which is caused by its low crystallization nucleation density, high brittleness, and poor toughness [6]. Therefore, the reinforcing of PHB has been frequently carried out, and natural fibers are always involved due to its biocompatibility and biodegradability [7]. In addition, as both nanocellulose and PHB are renewable materials, PHB-based biodegradable nanocomposites have great application prospects in the packaging field.

Nanocellulose is used as a nanotoughening agent to improve the toughness of polymers because of its large specific surface area, high reactivity, good dispersion in the polymer matrix, easy formation of hydrogen bonds between the nanocellulose and the polymer, and ability to reduce the crystallinity of the polymer [8]. Until now, researchers have focused on the preparation and properties of nanocellulose/bio-based composite materials, such as starch [9], polyvinyl alcohol (PVA) [10], waterborne polyurethane (WPU) [11], polycaprolactone (PCL) [12], polylactic acid (PLA) [13], and epoxy resin (EP) [14]. Studies have shown that the mechanical properties of nanocellulose/bio-based composites can be better improved than for other reinforcing materials such as nanoclay or carbon-based materials. 

The main methods of preparing nanocellulose are acidolysis and physical methods. Physical methods such as high-pressure homogenization [15], grinding/refining [16], and ultrasonic treatment [17] have been used to extract cellulosic fibers from wood fibers (WF) and plant fibers (PF), microcrystalline cellulose (MCC), cysts, algae, and bacteria. These methods use high shear force to expand and separate cellulose molecules into nanometer-sized microfibrils, which are called cellulose nanofibrils (CNFs) [18]. Acid hydrolysis is mainly used to extract crystalline particles from various kinds of cellulose (PF, WF, MCC, cysts, algae, and bacteria). The nanocellulose prepared by hydrolyzing amorphous areas in cellulose microfibers is called cellulose nanocrystals (CNCs) [19,20]. It has already been utilized for many potential applications such as paper, newspaper, textiles fibers, paper board, construction, etc. [21]. Some researchers have extracted CNCs as nanoreinforcement into bionanocomposite for biomedical and other value-added products in industrial applications [22]. Until now, two kinds of nanocellulose have been effectively used during the processing of reinforced PHA composites. Srithep et al. [23] prepared poly(3-hydroxybutyrate-co-3-hydroxyvalerate) (PHBV)/CNFs composites by liquid ammonia cooling and melt blending or by blending PHBV solution with a suspension of CNFs. Studies have shown that the addition of CNFs increases the Young’s modulus of PHBV/CNFs nanocomposites by nearly two times, and the tensile strength is slightly enhanced, but the elongation at break is reduced because of the agglomeration of CNFs in the PHBV matrix and the degradation of some PHBV. Cherpinski et al. [24] prepared a PHB/CNF/PHB three-layer composite film by an electrospinning coating technique and reported that in comparison with pure PHB, the elongation at break of the multilayer film was increased by 135.7%, and the Young’s modulus and tensile strength were decreased by 11.5% and 35.6%, respectively. This was due to the good adhesion property of the PHB film to CNF, which resulted in a multilayer film with a higher elongation at break than the pure PHB film. The decrease in tensile strength was probably related to the delamination failure during the fracture process. Seoane et al. [25] studied the effect of CNCs on PHB characteristics and stated that at 6% CNC concentration, the tensile modulus and strength of the nanocomposites were increased about 50% and 35%, respectively.

Nanocellulose with different morphologies has different effects on PHB performance. At present, the research on the improvement of PHB performance with nanocellulose only considers a single morphology (CNCs or CNFs) of nanocellulose and does not systematically compare the effects of the two morphologies on PHB performance. Therefore, in this paper, CNCs and CNFs were used as nanotougheners to blend PHB with nanocellulose. The effects of different contents of CNCs and CNFs on the mechanical properties, thermal properties, crystallization process, crystal structure, and surface morphology of PHB were systematically studied and compared. In addition, the process optimization of PHB toughening by nanocellulose and its mechanism was also evaluated in this paper.

## 2. Materials and Methods

### 2.1. Materials

Bleached pulp board was obtained from Guangxi Guitang Co., Ltd. (Nanning, China), sulfuric acid (98%, analytical grade) was obtained from Linyi Mengyang Chemical Co., Ltd. (Linyi, China), acetone was obtained from Yangzhou Tianda Chemical Co., Ltd. (Yangzhou China), Phosphotungstic acid was obtained from Xi’an Tianmao Chemical Co., Ltd. (Xi’an, China), while PHB powder (model: ENMAT Y3000P) and chloroform (analytical grade) were purchased from Ningbo Tianan Biomaterial Co., Ltd. (Ningbo, China), and Shanghai Honghe Industrial Co., Ltd. (Shanghai, China), respectively.

### 2.2. Preparation of Cellulose Nanocrystals (CNCs)

First, 10 g slurry of was weighed in a beaker to which 200 g (64 wt %) sulfuric acid was added while keeping the beaker in an ice bath. After complete dissolving, the beaker was taken out from the ice bath and hydrolyzed at room temperature for 3 h at a stirring speed of 1000 r/min, followed by dilution with 1000 ml of deionized water to terminate the reaction. The diluted dispersion was centrifuged for 15 min in a high-speed refrigerated centrifuge (Himac CR21N, Tokyo, Japan) until the pH of the dispersion was 7. Then, the sample was subjected to ultrasonic treatment for 5 min to obtain a uniform nanocellulose suspension (The experimental chart of the best choice of acid hydrolysis temperature, acid hydrolysis time, and acid hydrolysis concentration can be found in the Appendix A).

### 2.3. Preparation of Cellulose Nanofibers (CNFs)

A quantity of 200 g of dry pulp was soaked in distilled water until the slurry became soft. After balancing the water for 24 h, water was added to the slurry at a rate of 2% and soaked for 2 h. The soaked slurry was ground 5 to 6 times using an ultrafine pulverizer (MKZA10-15J, Tokyo, Japan). Distilled water was added to continue the swelling sufficiently, and the concentration was controlled at 0.5%. Then, the slurry dispersion was transferred to a high-pressure homogenizer (M-110EH30, Westwood, MA, USA) for 15 cycles to obtain a uniform nanocellulose suspension.

### 2.4. Preparation of Nanocellulose Chloroform Suspension

The obtained nanocellulose suspension was repeatedly centrifuged at a rotation speed of 12,000 r/min for 45 min at 10 °C. Acetone (3 mL) was added to the centrifuge tube to displace the nanofiber and centrifuged again at a rotation speed of 12000 r/min and at 10 °C for 15 min. To obtain an acetone suspension, 3 mL of chloroform was added to the centrifuge and centrifuged three times at a rotation speed of 12,000 r/min at 10 °C for 15 min to replace the acetone in the nanocellulose. Finally, the chloroform suspension of nanocellulose was obtained by stirring with a shear dispersing emulsifier for 5 min.

### 2.5. Preparation of Nanocellulose/PHB Composite Film

The PHB (2 g) was dissolved in chloroform (50 mL) at 90 °C and mechanically stirred at a speed of 500 rpm under reflux for 3 h. After completely dissolving, the nanocellulose suspension was added, stirred at a temperature of 60 °C, a rotation speed of 1000 r/min for 30 min, and then coated on a 25 cm × 25 cm glass plate by automatic film application machine (ZEHNTNER, ZAA2300, Basel, Switzerland). After the sample was volatilized by the solvent at room temperature, the residual solvent was further removed at 40 °C for 3 days. The composition of each component is presented in Table 1.

### 2.6. Characterization

#### 2.6.1. Scanning Electron Microscope (SEM) Analysis

The surface morphology of the film was observed by SEM (Philips Phenom^TM^, Amsterdam, Netherlands) at a voltage of 10 kV.

#### 2.6.2. Transmission Electron Microscopy (TEM) Analysis

The samples were observed by TEM (Gao Tech HT7700, Tokyo, Japan). The acceleration voltage was 80 kV. The samples were mixed with 0.1% CNCs or CNFs suspension. After magnetic stirring for 4 hours, ultrasound was used for 30 min. A drop of 0.1% nanocellulose suspension was dripped onto the copper net. In order to increase the contrast, the samples were dyed with 1% phosphotungstic acid aqueous solution for 5 min [26].

#### 2.6.3. Dynamic Light Scattering (DLS) Analysis

Dynamic light scattering (DLS) measurements were carried out on a Malvern Nanozetasizer (Zetasizer Nano ZSP, Malvern, United Kingdom). Samples were diluted in deionized water to 10^−2^ wt %. Five measurements were conducted from each sample, and their average was presented. All data were taken at 25 °C. For each sample, the final values represent an average of at least 5 measurements.

#### 2.6.4. X-Ray Diffraction (XRD) Test

The crystalline structure of the PHB and nanocellulose was analyzed using X-ray diffractometer (Rigaku MiniFlex600, Tokyo, Japan). All the experiments were conducted at a voltage of 40 kV and a current of 30 mA using nickel-filtered Cu–Kα radiation between 5 and 40°, at a rate of 5 °C/min.

#### 2.6.5. Thermal Characterization

The calorimetry analysis (DSC) was carried out by using a Nerzsch DSC instrument (Nerzsch, TA, Germany). A 10 mg sample was heated from room temperature to 200  °C at 10  °C/min under a nitrogen atmosphere for 3 min to eliminate the heat history, then cooled to −50  °C at 10  °C/min, kept for 3 min, and then again heated to 200  °C at 10  °C/min. Crystallinity of the composite film was calculated according to [27] using the following equation,
(1)$$X_{c}=\frac{\Delta H_{m}}{\Delta H_{m}^{0}\Delta w_{PHB}}\times 100\%$$
where $\Delta H_{m}$ is the measured value for PHB melting enthalpy, $\Delta H_{m}^{0}$ is the melting heat associated with pure crystalline PHB (146 J/mol) [28], and *w_PHB_* is the weight fraction of PHB in the composite.

The thermal stability of the films was analyzed using a synchronous thermogravimetric analyzer (TGA, JingKe ZRY-2P, Shanghai, China). The sample was placed in a crucible, and thermal analysis was performed under a nitrogen atmosphere in a temperature range of 20–600 °C at a heating rate of 10 °C/min.

#### 2.6.6. Mechanical Performance Test

According to ASTM D882-12, the film (50 ± 8 μm) was cut into a rectangle of 15 mm × 100 mm. The tensile strength, elongation at break, and Young’s modulus were measured using an electronic universal material testing machine (Instron 3367, Boston, MA, USA) equipped with a load cell of 1 kN. The initial clamping distance and testing speed were kept constant as 50 mm and 100 mm/min, respectively. Five parallel samples were used and the results were averaged.

#### 2.6.7. Barrier Performance Test

The oxygen transmission rates (OTRs) of the films were determined by the pressure method reported by Gali et al. [29], using an oxygen transmission instrument (Brugger GDP-C, Schramberg, Germany) with a low chamber vacuum and vacuum time of 5 min. Three parallel samples were used with a test area of 78.4 cm^2^ for each sample. The results of OTRs are presented as cm^3^/(m^2^·24 h).

The water vapor transmission rates of films were measured according to ASTM F1249 using a water vapor transmission instrument (MOCON PERMATRAN-W Model 3/61, Minneapolis, MN, USA) at a test temperature of 25 °C and a relative humidity of 75 %. Three parallel samples were used with a test area of 1 cm^2^ for each sample, and the results are presented as g/(m^2^·24 h).

#### 2.6.8. Optical Properties by UV–Vis Spectroscopy

Optical properties were measured at three different points from each sample using a UV–Vis spectrometer (Jena SPECORD plus 50, Jena, Germany) in the range of 400–700 nm. An average of the measurements was presented. 

#### 2.6.9. Atomic Force Microscopy (AFM) Test

The suspension of CNCs or CNFs was diluted with distilled water to 0.008 wt % and stirred for 4 hours. The suspension of 25 μL of nanocellulose was absorbed onto the silicon wafer with a pipette gun and dried naturally at room temperature. The silicon wafer was adhered to the iron wafer with double-sided adhesive. The AFM (Gao Tech 5100N, Tokyo, Japan) test was performed in tapping mode, and the cantilever elastic constant was 26 N/m.

## 3. Results and Discussion

### 3.1. Morphology of CNCs and CNFs

Figure 1 shows images of the two nanocelluloses. CNCs exhibited a "narrow, two-pointed" rod-like structure, while CNFs exhibited a grid-like structure, which were intertwined with each other. In addition, the diameters of both CNCs and CNFs were below 100 nm, which is in line with the existing size requirements for nanocellulose. The results obtained in our study were similar to those reported by predecessors [30].

Figure 2 shows the dispersion of bagasse fiber and two kinds of 1 wt % nanocellulose. The dispersion of bagasse fiber showed a clear phase separation, while the suspension of nanocellulose remained stable and did not show any phase separation or precipitation. This indicated a complete dispersion of nanocellulose and resulted in a colloidal structure. According to Dong et al. [31], sulfuric acid can induce the grafting of negatively charged sulfuric acid groups on the surface of nanocrystals during acid hydrolysis, which helps to stabilize colloidal suspensions through repulsive inter-particle forces. The CNFs prepared by the mechanical method form a colloid due to the strong shear, impact, and cavitation of bagasse bleached pulp under high pressure. Cavitation includes the formation, expansion, and implosion of microscopic bubbles when molecules in the liquid absorb ultrasonic energy. In the cavitation bubble and the surrounding area, there will be a violent shock wave, which can be used to separate the fiber from the cellulose fiber, so that the dispersion of bagasse fiber can be refined and the stable colloidal suspension can be obtained [17,32].

The DLS result showed a single peak with a hydrodynamic diameter of ∼60 nm, indicating the average size of the CNCs (Figure 3a). The CNFs suspension showed a nearly uniform particle size distribution with an average hydrodynamic diameter of ~106 nm (Figure 3b). Using the particle size analysis Nano Measurer drawing software, 50 nanocellulose from three different TEM images of CNCs and CNFs were selected, and the diameter and length of the nanocellulose were calculated. Figure 3c shows the diameter distribution of CNCs; their size distribution was from 2 to 18 nm, with the average diameter of 8.8 nm. Figure 3d shows the length distribution of CNCs, which ranged from 80 to 240 nm with an average length of 131.6 nm. Figure 3e shows the diameter distribution of CNFs, which were distributed between 6 and 26 nm with an average diameter of 12.6 nm. From the TEM diagram of CNFs, it can be seen that CNFs show a network structure at the microlevel, and the length range was identified. It can also be seen that the size of cellulose nanoparticles prepared by acid hydrolysis was clearly smaller than that of nanoparticles prepared by grinding and homogenization. The average size of the nanocellulose examined by TEM was about six or nine times smaller than that measured by DLS. Similar results were observed while extracting and characterizing nanocellulose from yerba mate sticks and wood pulp [33,34]. This discrepancy could be attributed to the dehydration of the nanocellulose during the sample preparation for TEM [35].

AFM can evaluate the dispersion performance of a nanomaterial and detect the surface morphology below the nanometer scale. Figure 4a–d shows examples of tapping mode AFM images from CNCs and CNFs particles deposited onto freshly cleaved mica surfaces. Despite a sonication of the suspensions before spreading on mica, some aggregates were observed on the surface. Interestingly, AFM results supported the TEM images of CNCs and CNFs, which showed that the CNCs have a rod-like structure while CNFs have a grid-like structure. Furthermore, the diameter of the two nanofibers prepared were below 100 nm.

### 3.2. Crystal Structure Analysis of CNCs and CNFs

The X-ray diffraction patterns of bagasse cellulose (BC), CNCs, and CNFs are shown in Figure 5. When the non-cellulose was removed and the amorphous regions were dissolved, the fibers showed an increased orientation along a specific axis. Typical cellulose I crystal diffraction peaks were 16.4°, 22.5°, and 34.5° at diffraction angles of 2θ [36]. Three diffraction peaks of the CNCs obtained by the acid treatment and the CNFs obtained by the mechanical method were at the same position, which corresponded to the characteristic absorption peaks of the crystal plane of the cellulose I-type crystal structure. This suggested that the acid and mechanical treatments of cellulose did not change the crystal form of cellulose. The diffraction peaks of the two nanocellulose at 22.5° were sharper, and the relative peak intensities were significantly (p ≤ 0.05) enhanced, which proves that the crystallinity has been increased. The possible reason for this increase in the crystallinity of cellulose by acid treatment could be the hydrogen ions that can enter into the amorphous area of cellulose and destroy the amorphous area [37]. Mechanical grinding and homogenization increased the crystallinity of cellulose due to the non-knotting portion of bagasse bleaching pulp being reduced correspondingly after the high-intensity shearing treatment in the preparation of nanocellulose and the more regular arrangement of fiber in the crystallization area [38,39].

### 3.3. Fracture Morphology Analysis of Nanocellulose/PHB Composite Film

Figure 6 shows the fracture morphology of PHB and PHB/nanocellulose composite films. The fracture of pure PHB was relatively smooth and had no concavity, but it did have convexity, which belongs to brittle fracture. When 1 wt % CNCs was added, the fracture surface of the composite become rough, and many evenly distributed holes appeared (circles in Figure 6b), which might be left by the CNCs when they were pulled out [40]. Compared with pure PHB, the nanocellulose prepared by the acid method was better wrapped by PHB. In addition, there was no obvious fiber nudity, and the agglomeration was less, which indicated that the interaction force between hydrophilic CNCs and hydrophobic PHB was stronger and the compatibility was better. This orientation of CNCs in the PHB matrix could be attributed to the adsorption of the PHB chain on the CNCs hydroxyl group at lower load [41]. Most of the nanocellulose was embedded on the polymer surface to provide more adhesive contact areas [42], which can enhance the mechanical properties of the PHB matrix and correspond to the test results of mechanical properties. When the content of CNCs was more than 1 wt %, the defects on the fracture surface expanded, especially the volume and number of holes, and the distribution of these defects became uneven (circles in Figure 6c). In contrast, the composite film shows brittle fracture. When the content of CNCs was increased to 5 wt %, the CNCs showed a serious agglomeration phenomenon, and the pore size and fracture surface size of CNCs increased sharply due to the strong intramolecular hydrogen bonds of the nanocellulose (circles in Figure 6d).

Figure 7 is a cross-sectional view of the different concentrations of cellulose nanocrystals added to the PHB/CNFs composite film. With the increase of CNFs content, the voids on the fracture surface changed from small to large, and the distribution becomes inhomogeneous. These holes were due to the pulling out of CNFs at the time of PHB/CNFs composite breakage in liquid nitrogen (Figure 7b–d arrow). At the same time, some blocks were observed, which were caused by agglomerated CNFs and single CNFs. CNFs’ agglomeration can lead to the weakening of the interface bonds between CNFs and PHB, and the composite film breaks near the agglomeration of CNFs during drawing, so it does not have its due toughness effect (see Figure 11).

### 3.4. Crystal Structure Analysis of Nanocellulose/PHB Composite Film

The XRD curves of pure PHB thin films, PHB/CNCs, and PHB/CNFs nanocomposite films with different nanocellulose contents are shown in Figure 8. The characteristic lattice planes of PHB were clearly observed in the diffraction pattern, and the 2θ were 13.8°, 16.5°, 20.3°, 22.3°, 25.1°, and 30.6°, respectively [43]. Among them, two strong scattering intensity peaks were detected near 2θ = 13.8° and 16.5°, which were designated as orthorhombic cells (020) and (110), respectively [44].

The weak reflection near 21.50° and 22.50° corresponded to (101) and (111). The stronger reflection near 25.1° and 30.6° corresponded to (121) and (200) [45]. It can be seen from the diagram that the main diffraction peaks of PHB/CNCs and PHB/CNFs composites were similar to those of pure PHB thin films, indicating that the addition of CNCs and CNFs has no effect on the crystal structure of PHB. Nanocellulose can be embedded in PHB crystals by hydrogen bonds, which leads to the shift of the XRD peak to a lower angle at 2θ = 13.5°. The crystallization degree of the composites was calculated by Jade software, as shown in Table 2. When the content of CNCs was 1 wt %, the crystallinity of the composite was lower than that of PHB, which indicates that CNCs can be better embedded into the PHB matrix at the lower content. This might be due to the intertwining of PHB macromolecular chains and CNFs. The rigid structure of nanocellulose restricts the movement of molecular chains in the crystallization process of PHB, resulting in the difficulty of crystallization and the decrease in crystallinity. When the content of CNCs was increased than 1 wt %, the main diffraction peak strengths of the nanocomposite were higher than those of pure PHB. This indicated that the crystallinity of the nanocomposites becomes larger at higher concentrations of CNCs. Furthermore, the 2θ of crystal planes (020) and (110) gradually becomes smaller, which was because CNCs and CNFs act as nucleating agents in the composite films. Increasing the number of nucleating agents and crystals in the system can increase the crystallinity. Our results were in agreement with the work of Ten et al. [46].

### 3.5. Melting and Non-Isothermal Crystallization Behavior of Nanocellulose/PHB Composite Film

Crystallinity is an important parameter that primarily affects the physical properties of biodegradable polymers. The non-isothermal crystallization behavior of the polymer is presented in in Figure 9 and Figure 10. Figure 9 shows the second DSC heat curve for pure PHB, PHB/CNCs, and PHB/CNFs composites. The thermal performance characteristic parameters including the melting temperature (*T*_m_), cold crystallization temperature (*T*_c_), melting enthalpy (Δ*H*_m_), and crystallinity of the material (*X*_c_) are summarized in Table 3.

From the secondary heat curve of DSC in Figure 9, it can be seen that with the addition of CNCs to PHB, the Tm increased from 168.2 °C to 171.0 °C, and with the increase of CNFs content from 1 wt % to 5 wt %, the *T*_m_ of PHB/CNCs composites increased from 168.2 °C to 172.5 °C. Interestingly, when the CNFs and CNCs were used at the same concentrations, the *T*_m_ of PHB/CNFs was higher than that of PHB/CNCs. This indicates that the nucleation of nanoscale CNFs was significantly stronger which further improved the integrity of PHB crystallization than that of CNCs. The toughness of PHB reduced by the improvement of the PHB crystal structure, which corresponded to the test results of mechanical properties.

Figure 10 shows the DSC first cooling curve of pure PHB, PHB/CNCs, and PHB/CNFs composite films. It can be seen from Figure 10 and Table 3 that when 1 wt % CNCs was added to PHB, the *T*_c_ of the composite film increased from 79.2 °C to 97.5 °C, which further indicated that the crystallization of PHB becomes easier. This increase in temperature was due to the addition of CNCs to PHB, which played the role of a heterogeneous nucleating agent, reduced the energy barrier of PHB nucleation, significantly increased the nucleation rate, and promoted melting crystallization. However, with the continued increase in CNCs content, the *T*_c_ of PHB/CNCs decreased from 97.5 °C to 86.8 °C; this decrease of 10.7 °C indicated that the nucleation ability of CNCs decreased, which was due to agglomeration of nanocellulose in the matrix. When CNFs were added into the PHB substrate, the *T*_c_ of the composite film increased from 79.2 °C to 94.8 °C and then decreased to 83.5 °C. When the contents of CNCs and CNFs in the PHB substrate were equal, the *T*_c_ of the CNCs/PHB composite film was higher than that of the CNFs/PHB composite film, which indicates that CNCs are more beneficial to the crystallization of PHB than CNFs.

### 3.6. Effect of Nanocellulose on the Mechanical Properties of PHB

A uniaxial tensile test was conducted to determine the effect of cellulose nanocellulose on the mechanical properties of PHB, as shown in Figure 11. Figure 11a–c show the tensile strength, elongation at break, and Young’s modulus of the composite films with different nanocellulose concentrations. The results showed that a low proportion of CNCs has a significant effect on the mechanical properties of PHB. At 1 wt % of CNCs, all the mechanical properties of the composite films were the best. The elongation at break increased from 3.40% to 6.5%, and Young’s modulus increased from 1.4 GPa to 1.7 GPa, which represents an increase of 91.2% and 18.4%, respectively, compared with pure PHB. On the other hand, the tensile strength decreased from 31.2 MPa to 27.1 MPa, which represents a decrease of 3.5%. The Young’s modulus of CNCs/PHB composite films increased from 1.7 GPa to 2.0 GPa, which was 17.5% higher than that of pure PHB. However, the tensile strength and elongation at break of the composite film decreased to 22.8 MPa and 2.6%, respectively. This is because of the CNCs contents, which can make the molecular arrangement more compact, reduce the free volume, and increase the elongation at break. As shown in Figure 6, the CNCs at low concentration uniformly dispersed into PHB, and the surface defects were relatively low. From the perspective of crystallization behavior, CNCs played the role of a heterogeneous nucleating agent in PHB, accelerated the crystallization process, reduced the crystallization degree of PHB, and basically solved the problem of poor toughness of PHB (as shown in Table 3). When the concentration of CNCs increased from 1 wt %, the tensile strength and elongation at break of the composite films decreased, but the Young’s modulus continuously increased. This was mainly due to the fact that CNCs were derived from natural biomass with an organic rigid material having a stiffness of 140–220 GPa [47,48]. The dispersion of CNCs can reach about 1 wt % by the solvent exchange method; however, the agglomeration of CNCs can be observed under a higher load of CNCs, and their effect on the mechanical properties of PHB decreases gradually. At this time, the interfacial binding force between hydrophilic CNCs and the hydrophobic polymer decreases, and the intermolecular hydrogen bond binding force increases [49]. Other researchers [50] stated that under higher CNCs loading, CNCs begin to aggregate in the form of clusters on the surface of the polymer, and the side effects on mechanical properties are more significant.

When the concentrations of CNFs increased from 1 wt % to 5 wt %, the tensile strength and elongation at break of PHB continuously decreased; the tensile strength decreased from 31.2 MPa to 10.6 MPa, and the elongation at break decreased from 3.4% to 1.4%, which was 66.0% and 58.8% lower than that of pure PHB, respectively. However, Young’s modulus increased from 1.7 GPa to 1.97 GPa, which was 15.9% higher than that of pure PHB. This was due to the agglomeration of CNFs in the PHB matrix, which was consistent with the SEM diagram of CNFs/PHB (Figure 6).

### 3.7. Barrier Properties of Nanocellulose/PHB Composite Film

The shelf life, safety, and health of packaged food are strongly affected by the humidity in the package, because humidity is one of the important factors that provide a suitable environment for the growth of unnecessary microorganisms. The water vapor and oxygen permeability of composite films can be reduced by incorporating nanoparticles into the polymer matrix. Therefore, the water vapor and oxygen transmission rates of pure PHB, PHB/CNCs, and PHB/CNFs nanocomposite films were investigated. The results are shown in Figure 12 and Figure 13.

The water vapor and oxygen transmission rates of PHB were significantly reduced with the addition of the CNCs nanoparticles. The water vapor and oxygen transmittance of pure PHB membrane were 298.7 and 301.2 cm^3^/(m^2^·24 h), respectively. With the addition of 1% CNCs, the water vapor and oxygen transmission rates greatly decreased to 157.9 and 140.4 cm^3^/(m^2^·24 h), representing a decrease of 52.9% and 46.6%, respectively. When the CNCs concentration increased to 3% CNCs, the water vapor and oxygen transmittance rates of the composite membrane decreased to 85.2 g/(m^2^·24 h) and 87.1 g/(m^2^·24 h), respectively. This indicates that the resistance of the composite film to oxygen molecular penetration has increased by the nanometer size effect of CNCs. The rheological study of Bharadwaj et al. [51] shows that the instantaneous penetration network of CNCs formed in the polymer matrix by hydrogen bonds increases the resistance of CNCs’ penetration to gas permeability. The increased permeability resistance provided by the CNCs’ percolation network in the polymer matrix might be the cause of this situation [52]. Furthermore, the embedding of CNCs into the PHB matrix might also be the reason for the decrease in oxygen transmittance rate. When the content of CNCs was increased to 5%, the water vapor transmittance and oxygen transmittance increased to 120.4 g/(m^2^·24 h) and 101.5 cm^3^/(m^2^·24 h), respectively. This was due to CNCs’ intercalation effect and the osmotic network that can be ignored at a higher CNCs content, which leads to a higher oxygen transmittance rate. This might also be due to weak dispersion and agglomeration of CNCs nanoparticles.

Water vapor and oxygen transmittance rates of PHB also significantly (p ≤ 0.05) decreased with the addition of CNFs nanoparticles. The water vapor transmittance of the composite film decreased from 298.7 to 140.4 g/(m^2^·24 h) and oxygen transmittance decreased from 301.2 to 125.8 g/(m^2^·24 h), representing a decrease of 47.0% and 41.8%, respectively. This effect was similar as found in the addition of CNCs. Due to the hydrogen bonding between CNFs and PHB in the composite film, a physical crosslinked network was formed in the system, where the gas dissolution process becomes more difficult. At this time, according to the tortuous path theory of composites, water molecules have to go through more tortuous paths in nanocomposites; therefore, the addition of CNFs improves the water vapor barrier performance of composite membranes. At 5% CNFs content, the water vapor and oxygen transmittance rates of the composite membrane increased to 198.6 and 177.2 cm^3^/(m^2^ 24 h), respectively, and this increase was greater than that of CNCs. On the one hand, due to the existence of gaps in the cross-structure of CNFs, under the action of pressure, gas molecules can pass through this gap; hence, the agglomeration of CNFs with the same addition amount was more obvious.

### 3.8. Transparency of Nanocellulose/PHB Composite Films

Transmittance is one of the indexes to characterize the transparency of materials, and it is also one of the means to judge the compatibility of composite films. The transparency of the nanocomposite film is determined by UV–visible spectroscopy, and the spectra are shown in Figure 14. The specific values of the transmittance of nanocellulose/PHB composite films at the main wavelengths of visible light, at different nanocellulose mass fractions, are summarized in Table 4.

With the increase in wavelength, the transmittance of the film is lower than that of pure PHB films except for 1% PHB/CNCs samples (Figure 14). With the addition of 1% nanocellulose, the transmittance of the composite film slightly increased. This might be uniformly dispersion of low nanocellulose in the PHB matrix. Another possible explanation could be the size of nanocellulose, which is smaller than the wavelength of visible light, and therefore the transmission of visible light did not affect it.

As presented in Table 4, the transmittance of the composite film decreased with the increase in nanocellulose concentration. This was probably due to the poor compatibility between hydrophilic nanocellulose and hydrophobic PHB, as well as the fact that the aggregation of nanocellulose in the PHB matrix was not conducive to light transmission; hence, the transmittance of the film began to decline.

### 3.9. Thermal Stability of Nanocellulose/PHB Composite Films

Figure 15 shows the thermogravimetric (TG) and differential thermogravimetric (DTG) curves of PHB/CNCs composite films. The main mass loss range of the PHB/CNCs composite film was 250–300 °C, and the whole process shows a "Z" font. The initial decomposition temperature (*T*_0_) of pure PHB was 254 °C, and that of the composite films was almost the same (Table 5). With the increase of CNCs concentration, the peak of the fastest decomposition temperature (*T*_max_) of the PHB/CNCs composite film was shifted to the right, from 286.5 °C to 291.7 °C. This shows that the addition of a small amount of CNCs does not significantly improve the thermal performance of PHB. This might be due to the strong interaction between CNCs and PHB, which changed the thermal stability of PHB. This might also be due to the formation of hydrogen bonds between the carbonyl group of PHB and the hydroxyl group of CNCs, which slowed down the random breaking chain of PHB, thus improving the thermal stability of PHB [42].

Figure 16 shows the TG and DTG curves of PHB/CNFs composite films. As can be seen from Figure 16a, the main thermal weight loss region and the whole process of PHB/CNFs were consistent with those of PHB/CNCs composite films. Figure 16b shows the DTG curves of PHB/CNFs composite thin films. PHB/CNFs composite films also showed single-peak weightlessness, which was similar to the process of PHB/CNCs composite films.

The initial decomposition temperature (*T*_0_) of the composite films was not affected by CNFs (Table 5). With the increase in CNFs content, the peak of the highest *T*_max_ of PHB/CNF composite films shifted to the right from 286.5 to 294.2 °C. Compared with the thermal stability of the two kinds of nanocomposite films, it was found that CNFs were more favorable than CNCs in improving the thermal stability of PHB. This could be because the reticular cross-structure formed by CNFs at the microlevel can inhibit the degradation of PHB [53].

## 4. Conclusions

In this paper, the effects of the morphology and content of nanocellulose on the properties of PHB were systematically compared. It was concluded that the morphology of bagasse nanocellulose was rod-shaped, and the toughening effect of bagasse nanocellulose on PHB was the highest at 1 wt %. Compared with pure PHB, the elongation at break and Young’s modulus of CNCs/PHB composite film increased by 91.2% and 18.4%, respectively, while the tensile strength decreased by 3.5%. With the increase in CNCs content, the elongation at break and *T*_c_ of CNCs/PHB composite films first increased and then decreased, while the crystallinity and gas permeability first decreased and then increased. With the addition of CNCs, the tensile strength and light transmittance decreased, and the Young’s modulus, *T*_m_, and *T*_max_ increased. Similarly, with the increase in CNFs content, the *T*_c_ of CNFs/PHB composite films first increased and then decreased while the barrier properties initially decreased and then increased. The elongation at break, tensile strength, and light transmittance were reduced, and the Young’s modulus, crystallinity, *T*_m_, and *T*_max_ increased.

At 1 wt % CNCs content, the toughness of PHB increased; as in the crystallization behavior, CNCs act as heterogeneous nucleating agents in PHB, thereby reducing the crystallinity of PHB. In the aspect of interfacial bonding, CNCs were bound to PHB by hydrogen bonding. In addition, at 1 wt % CNCs content, most of the nanocellulose was embedded on the surface of the polymer, which provided more bonding contact areas, and hence the interface compatibility between CNCs and PHB was better.

The bio-based nanocomposites prepared in this study are expected to replace petroleum-based non-degradable materials in the packaging field. In addition, the application of agricultural and forestry by-products has been broadened.

## Figures and Tables

**Figure 1 polymers-11-02063-f001:**
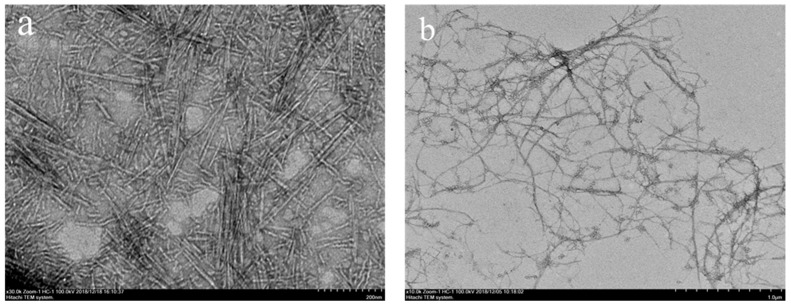
TEM images of the two nanocelluloses: (**a**) cellulose nanocrystals (CNCs); (**b**) cellulose nanofibers (CNFs).

**Figure 2 polymers-11-02063-f002:**
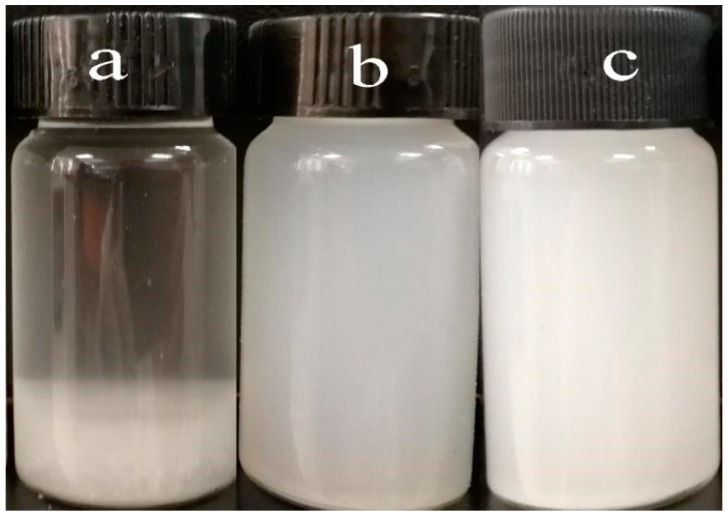
Appearance of bagasse fibers and two nanocelluloses: (**a**) Dispersion of bagasse fibers; (**b**) Suspension of CNCs; (**c**) Suspension of CNFs.

**Figure 3 polymers-11-02063-f003:**
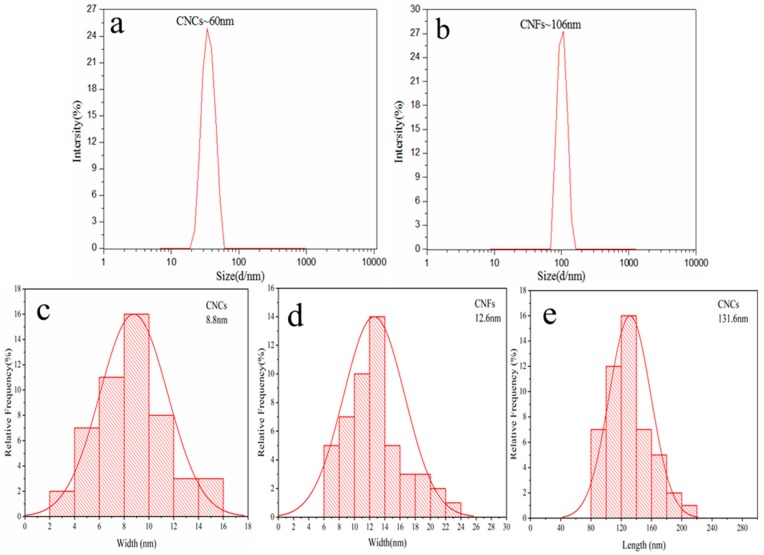
(**a**) Dynamic light scattering (DLS) measurement of the cellulose nanocrystals (CNCs) suspension, (**b**) DLS measurement of the cellulose nanofibers (CNFs) suspension, (**c**) TEM measurement of the CNCs—width, (**d**) TEM measurement of the CNCs—length, (**e**) TEM measurement of the CNFs—width.

**Figure 4 polymers-11-02063-f004:**
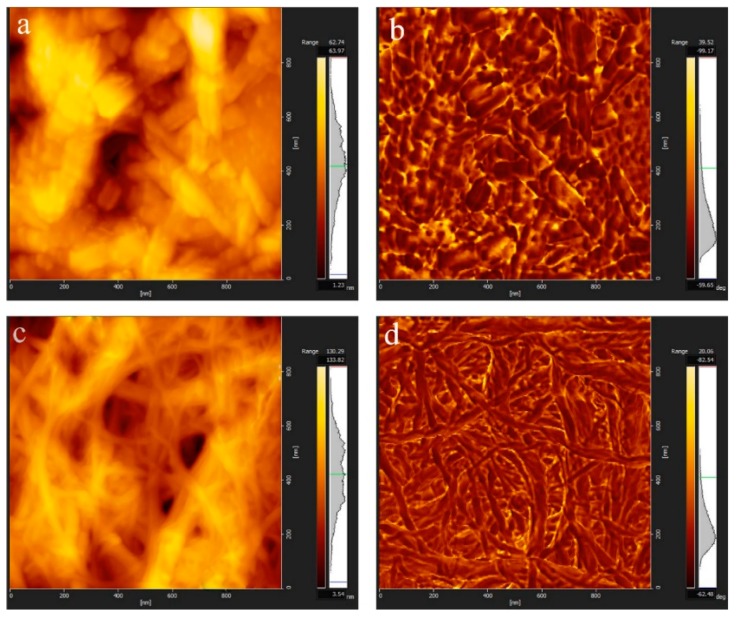
Atomic force microscopy (AFM) diagrams of the two nanocelluloses: (**a**) topography of CNCs, (**b**) phase diagram of CNCs, (**c**) topography of CNFs, and (**d**) phase diagram of CNFs.

**Figure 5 polymers-11-02063-f005:**
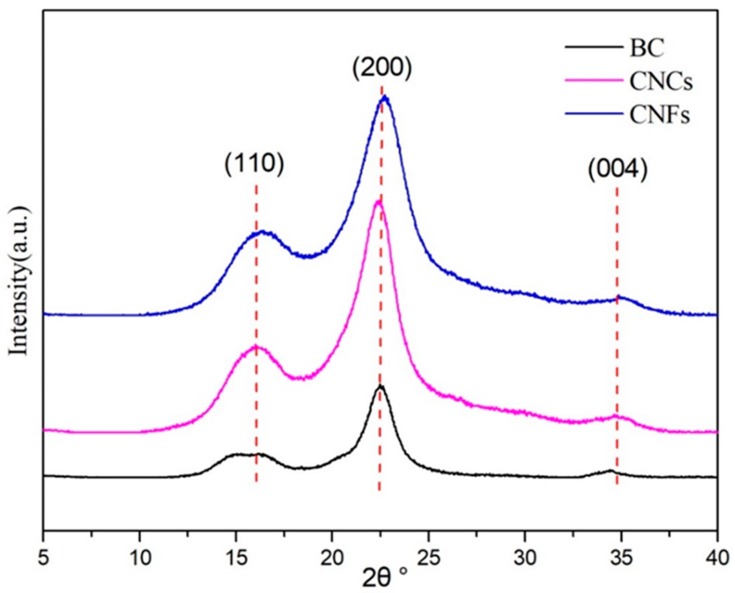
X-ray diffraction pattern of bagasse cellulose (BC), CNCs, and CNFs.

**Figure 6 polymers-11-02063-f006:**
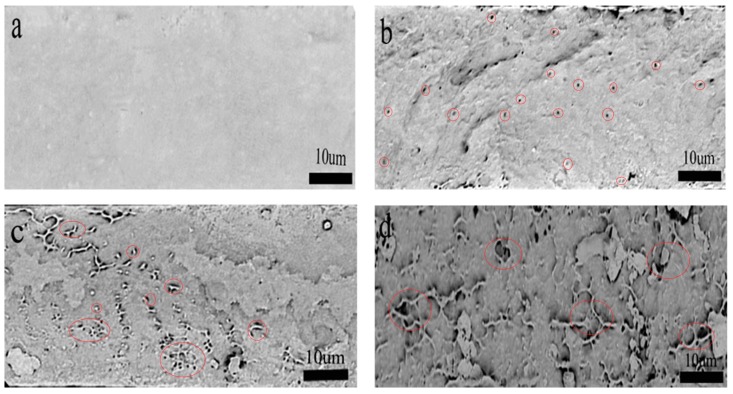
Effect of the different concentrations of cellulose nanocrystals on the impact cross-section of PHB/CNCs (SEM image): (**a**) pure PHB, (**b**) 1 wt %, (**c**) 3 wt %, (**d**) 5 wt %.

**Figure 7 polymers-11-02063-f007:**
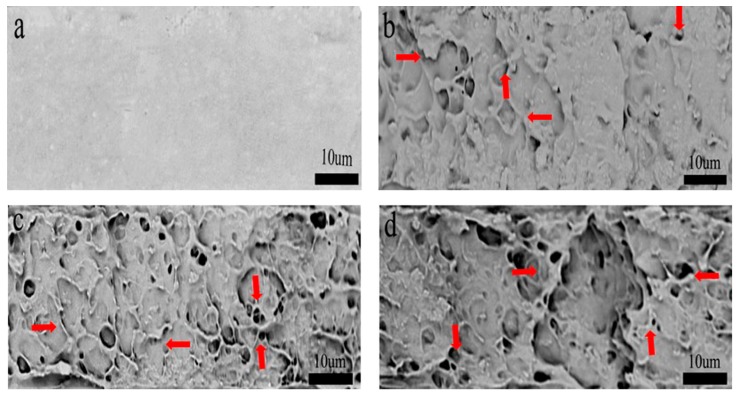
Effect of the different concentrations of cellulose nanofibrils on the impact cross-section of PHB/CNFs (SEM image): (**a**) pure PHB, (**b**) 1 wt %, (**c**) 3 wt %, (**d**) 5 wt %.

**Figure 8 polymers-11-02063-f008:**
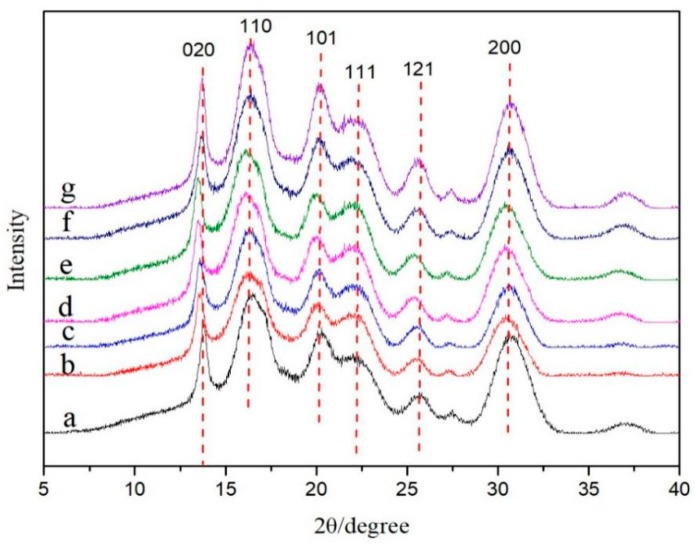
X-ray diffraction (XRD) patterns of PHB/CNCs and PHB/CNFs composite films: (**a**) PHB; (**b**) PHB/CNC1; (**c**) PHB/CNC3; (**d**) PHB/CNF5; (**e**) PHB/CNF1; (**f**) PHB/CNF3; (**g**) PHB/CNF5.

**Figure 9 polymers-11-02063-f009:**
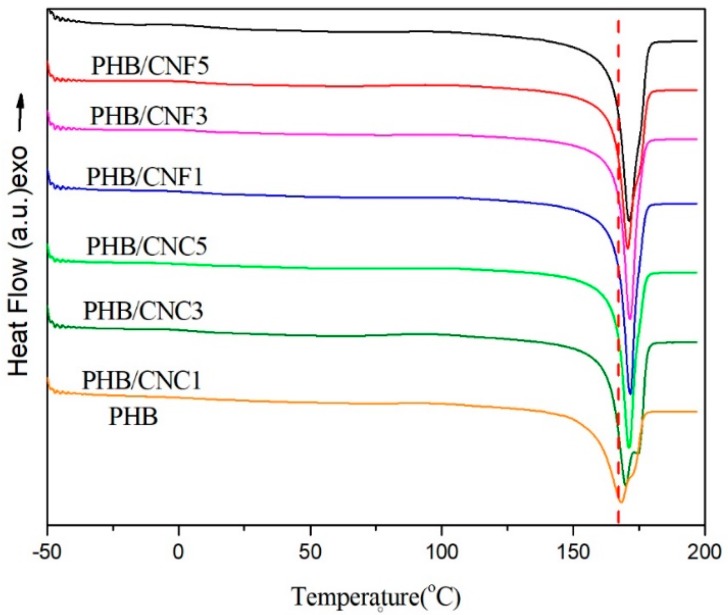
DSC second heating curve of pure PHB, PHB/CNCs, and PHB/CNFs composite films.

**Figure 10 polymers-11-02063-f010:**
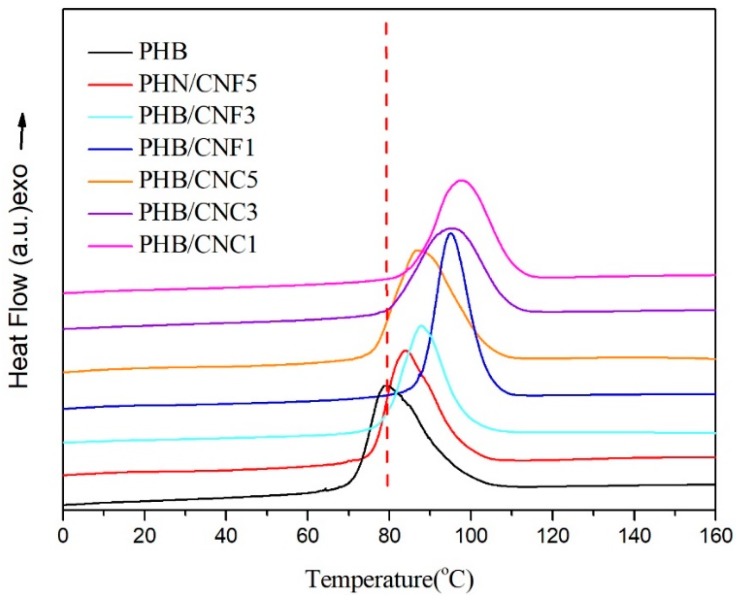
DSC first cooling curve for pure PHB, PHB/CNCs, and PHB/CNFs composite films.

**Figure 11 polymers-11-02063-f011:**
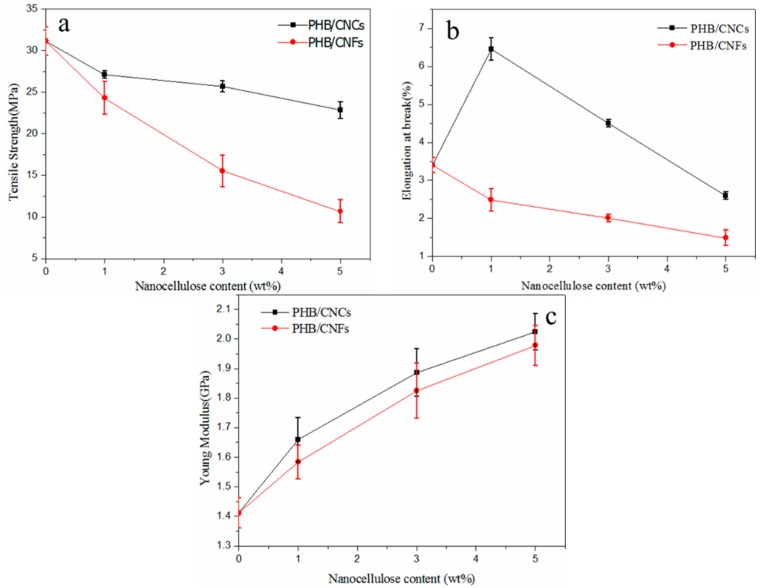
Effect of nanocellulose content on the mechanical properties of composites: (**a**) tensile strength, (**b**) elongation at break, and (**c**) Young’s modulus.

**Figure 12 polymers-11-02063-f012:**
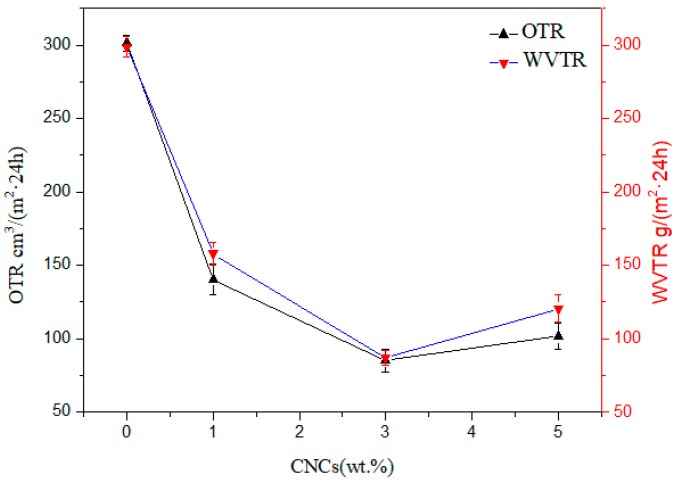
Water vapor and oxygen transmission rates against CNCs content for neat PHB and PHB/CNCs nanocomposite film.

**Figure 13 polymers-11-02063-f013:**
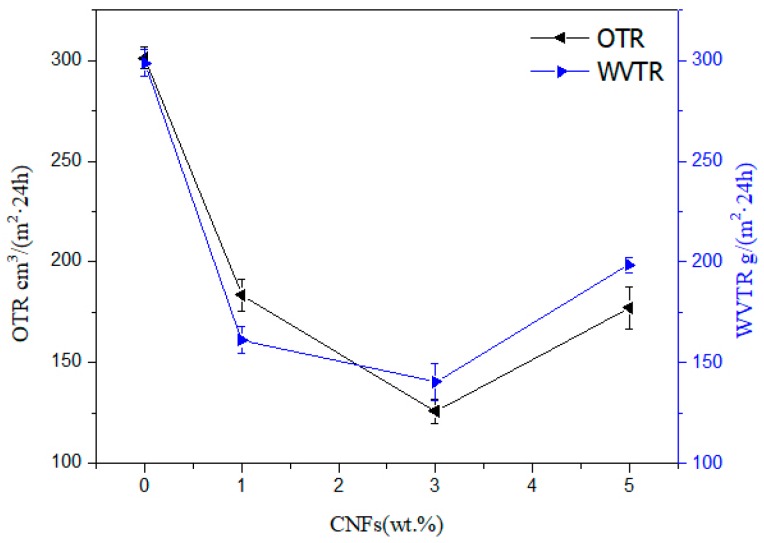
Water vapor and oxygen transmission rate against CNFs content for neat PHB and PHB/CNFs nanocomposite film.

**Figure 14 polymers-11-02063-f014:**
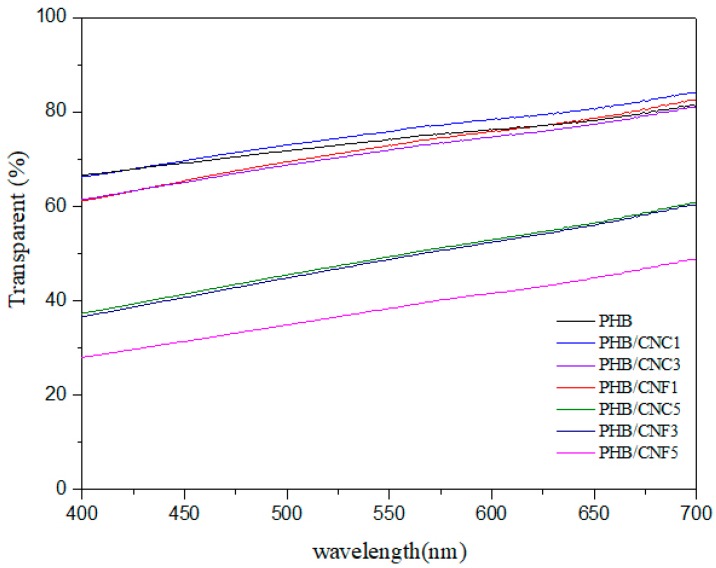
Transmittance curves of nanocellulose/PHB nanocomposite films with UV–visible light at different nanocellulose mass fractions.

**Figure 15 polymers-11-02063-f015:**
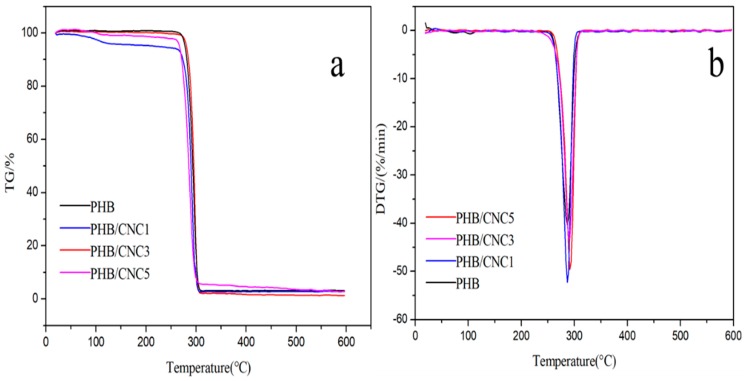
(**a**) Thermogravimetric (TG) and (**b**) differential thermogravimetric (DTG) curves of PHB/CNCs composite films.

**Figure 16 polymers-11-02063-f016:**
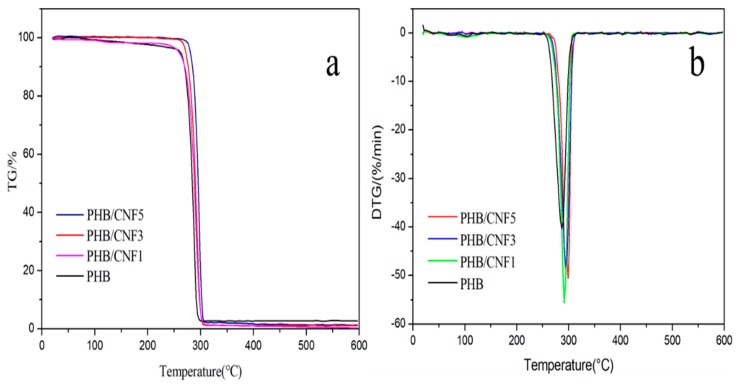
(**a**) TG and (**b**) DTG curves of PHB/CNCs composite films.

**Table 1 polymers-11-02063-t001:** Mass ratio of nanocellulose/polyhydroxybutyrate (PHB) composites.

Sample Serial Number	PHB (wt %)	Percentage of Absolute Dry Mass and PHB Mass of Nanocellulose (wt %)
PHB	100	0
PHBCNC1	100	1
PHBCNC3	100	3
PHBCNC5	100	5
PHBCNF1	100	1
PHBCNF3	100	3
PHBCNF5	100	5

**Table 2 polymers-11-02063-t002:** Crystallinity of nanocellulose/PHB composite film.

Sample	Crystallinity (%)
PHB	56.61
PHB/CNC1	54.6
PHB/CNC3	56.85
PHB/CNC5	59.37
PHB/CNF1	57.9
PHB/CNF3	59.5
PHB/CNF5	60.1

**Table 3 polymers-11-02063-t003:** Melting parameters of PHB, PHB/CNCs, and PHB/CNFs composite films.

Sample	Second Melting			First Melting
	*T*_m_/°C	Δ*H*_m_/(J/g)	*X*_c_/%	*T*_c_ /°C
PHB	168.2	94.6	64.8	79.2
PHB/CNC1	169.5	88.3	61.1	97.5
PHB/CNC3	170.9	90.8	64.2	96.1
PHB/CNC5	171.0	93.1	67.1	86.8
PHB/CNF1	170.8	91.2	63.1	94.8
PHB/CNF3	171.3	94.8	66.9	87.5
PHB/CNF5	172.5	96,8	69.8	83.5

**Table 4 polymers-11-02063-t004:** Transmittance of PHB, PHB/CNCs, and PHB/CNFs composite films.

Sample	Transmittance (%)			
	400 nm	500 nm	600 nm	700 nm
PHB	66.69	71.84	76.25	82.56
PHB/CNC1	66.3	73.04	78.43	84.14
PHB/CNC3	61.46	69.47	75.83	81.42
PHB/CNC5	37.36	45.52	52.96	60.85
PHB/CNF1	61.2	68.76	74.71	81.06
PHB/CNF3	36.69	44.85	52.45	60.44
PHB/CNF5	28.08	34.93	41.6	48.88

**Table 5 polymers-11-02063-t005:** PHB, PHB/CNCs and PHB/CNFs thermogravimetric data.

Sample	*T*_0_/°C	*T*_max_/°C	600 °C Residual Ash (%)
PHB	254.0	286.5	1.12
PHB/CNC1	254.1	288.4	0.97
PHB/CNC3	250.7	290.2	1.01
PHB/CNC5	252.7	291.7	1.76
PHB/CNF1	254.4	291.4	2.42
PHB/CNF3	255.2	294.6	2.86
PHB/CNF5	255.6	298.2	1.17

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
