# Peer review of "Effects of Cellulose Nanocrystals and Cellulose Nanofibers on the Structure and Properties of Polyhydroxybutyrate Nanocomposites"

_polymers, 2019, doi:10.3390/polym11122063_

Round 1
Reviewer 1 Report
Authors have prepared polyhydroxybutyrate (PHB)-based nanocomposites reinforced with cellulose nanocrystals and nanofibres (CNCs and CNFs). This study is interesting and suitable to be published in this journal. However, there are some points to be resolved and improved before consideration.
In Introduction section, the description about bagasse-based cellulose nanocrystals and/or nanofibres should be incorporated in terms of preparation and use in nanocomposites. Author may refer following articles: Journal of Materials Physics and Chemistry, 2(1), 2014, 1-8; Polym. Compos, 2(3), 2014, 23-27. In section 2.2, reviewer is afraid if CNCs can be extracted sufficiently using 64% sulfuric acid at room temperature for 3 h. Is this optimized process or referred to any article?. For example, authors may refer following article based on CNCs preparation: Journal of Materials Physics and Chemistry, 2(1), 2014 and Carbohydrate polymers, 193, 2018, 228-238. In section 2.5, how authors controlled film thickness in all samples? As in section 2.6.1 including throughout the manuscript, authors have described antimicrobial films, are these films antimicrobial? If yes, then how? Did authors added any agent in it and performed any experiment for it? Please mention the thickness of films in Section 2.6.5. In Figure 1, size dimensions of CNCs seem to be different than Figure 2. Lengths of CNCs are in higher value as in SEM image. How authors calculated these dimensions, please describe it in the manuscript? In Figure 7, clear peaks of CNCs or CNFs are not observed. Could authors provide XRD of CNCs or CNFs in addition to this data and describe why this is?, so that readers may have better insight. As authors have described in morphological discussion, some holes or volumes on fracture surface of films based on load transfer within the films are observed. These defects might have significant impact on mechanical properties during tensile strain. Therefore, authors should add actual plots of mechanical testing (may be provided in Supporting information), if possible, so that clear behavior of the nanocomposite films could be seen. Authors have described that thermal properties of films were improved with increased CNCs content, but in Figure 14 it is not evident. In this Figure, PHB/CNCs1 and PHB/CNCs5 show decreased thermal degradation than PHB and PHB/CNCs3. May there be an effect of sulfated-CNCs (as prepared in this study) on properties of films? For this, authors may refer following article: Journal of Materials Physics and Chemistry, 2(1), 2014.
Author Response
Date: December 3, 2019
Manuscript ID: polymers-660401
Title: " Effects of Cellulose Nanocrystals and Cellulose Nanofibers on the Structure and Properties of Polyhydroxybutyrate Nanocomposites "
Dear reviewer,
Thank you for your comments. We find those comments very valuable and helpful, and have taken them into full consideration in our revised manuscript. We have revised the manuscript by adding supporting experiments and related explanations. The main revisions marked in red in the revised manuscript. The point by point responses to the reviewers’ comments are listed below this letter.
With best wishes,
Yours sincerely,
Bobo Zhang, Chongxing Huang and Hui Zhao.
Authors have prepared polyhydroxybutyrate (PHB)-based nanocomposites reinforced with cellulose nanocrystals and nanofibres (CNCs and CNFs). This study is interesting and suitable to be published in this journal. However, there are some points to be resolved and improved before consideration.
Comment 1: In Introduction section, the description about bagasse-based cellulose nanocrystals and/or nanofibres should be incorporated in terms of preparation and use in nanocomposites. Author may refer following articles: Journal of Materials Physics and Chemistry, 2(1), 2014, 1-8; Polym. Compos, 2(3), 2014, 23-27.
Response: Response: Thanks for the comment. In the revision, We quote two sentences from the two references (Materials Physics and Chemistry, 2(1), 2014, 1-8; Polym. Compos, 2(3), 2014, 23-27) you provided to incorporate in terms of preparation and use in nanocomposite. The details are as follows:
1. It has already been utilized for many potential applications such as paper, newspaper, textiles fibers, paper board, construction, etc. [Journal of Materials Physics and Chemistry, 2(1), 2014, 1-8].
2. Some researchers have extracted CNCs as nanoreinforcement into bionanocomposite for biomedical and other value-added products in industrial applications [Polym. Compos, 2(3), 2014, 23-27].
Comment 2: In section 2.2, reviewer is afraid if CNCs can be extracted sufficiently using 64% sulfuric acid at room temperature for 3 h. Is this optimized process or referred to any article? For example, authors may refer following article based on CNCs preparation: Journal of Materials Physics and Chemistry, 2(1), 2014 and Carbohydrate polymers, 193, 2018, 228-238.
Response: Thanks for the kind advice. We follow the steps from many literatures (Cellulose, 2009, 16(3), 455-465; Cellulose, 2006, 13(2), 171-180; Carbohydrate Polymers, 2016, 157, 1821-1829; International Journal of Biological Macromolecules, 2018, 106, 433-446; Industrial Crops and Products, 2016, 93,48–57 etc.) to prepare nanocellulose and find that the yield of nanocellulose is very low and the size does not meet the requirements of subsequent experiments. The experimental data for the optimal process conditions for preparing nanocellulose has been added in Figure S1. Experimental images are provided in the supporting information.
Figure S1. Effects of sulfuric acid concentration, acid hydrolysis time, and acid hydrolysis temperature on the yield of CNCs.
From the Figure S1, the best preparation process of nanocellulose can be obtained. The sulfuric acid mass concentration is 64%. The acid hydrolysis time is 3 hours, and the acid hydrolysis temperature is 25 °C. At this time, the yield of nanocellulose is 38.5%.
Comment 3: In section 2.5, how authors controlled film thickness in all samples?
Response: As shown in the Figure S2, the automatic film application machine (ZEHNTNER, ZAA2300) was used to prepare the film, and thickness of the film by adjusting the stick height and the film speed on the film coating machine.
Figure S2. Machine stick on automatic film application machine.
Comment4: As in section 2.6.1 including throughout the manuscript, authors have described antimicrobial films, are these films antimicrobial? If yes, then how? Did authors added any agent in it and performed any experiment for it?
Response: Thanks for the detailed comment. We are sorry that it is a misquote. The film in this article is not an antibacterial film, which is not important for our conclusions. The “antimicrobial” mentioned in the article 4 times (Page3-line122, Page4-line148, Page4-line159, Page3-line163) has been deleted.
Comment 5: Please mention the thickness of films in Section 2.6.5.
Response: Thanks for the comment. The thickness of the films (50±8um) have been mentioned in section 2.6.6.
Comment 6: In Figure 1, size dimensions of CNCs seem to be different than Figure 2. Lengths of CNCs are in higher value as in SEM image. How authors calculated these dimensions, please describe it in the manuscript?
Response: We provide original TEM images of the two nanocelluloses in Figure 1. Figures 1 and 2 are data from the same batch of samples.
Figure 1. TEM images of the two nanocelluloses: (a) CNCs; (b) CNFs.
The diameter and length of nanocellulose were calculated by the particle size analysis Nano Measurer drawing software to select 50 nanocellulose from three different TEM images of CNCs and CNFs. The details has been added in Line 206-210, Page 6.
Comment 7: In Figure 7, clear peaks of CNCs or CNFs are not observed. Could authors provide XRD of CNCs or CNFs in addition to this data and describe why this is?, so that readers may have better insight.
Response: Thanks for the comment. We added a supporting experiment to test the XRD patterns of bagasse cellulose (BC), CNCs and CNFs, as shown in Figure 5.
Figure 5. X-ray diffraction pattern of BC, CNCs and CNFs.
The X-ray diffraction patterns of bagasse cellulose (BC), CNCs and CNFs are shown in Figure 5. When the non-cellulose was removed and the amorphous regions were dissolved, the fibers showed an increased orientation along a specific axis. Typical cellulose I crystal diffraction peaks were 16.4°, 22.5° and 34.5° at diffraction angles 2θ. Three diffraction peaks of the CNCs obtained by the acid treatment and the CNFs obtained by the mechanical method were at the same position, corresponded to the characteristic absorption peaks of the crystal plane of the cellulose I-type crystal structure. This suggested that the acid and mechanical treatments of cellulose did not change the crystal form of cellulose. The diffraction peaks of the two nanocellulose at 22.5° were sharper, and the relative peak intensities were significantly (p≤ 0.05) enhanced, which proves that the crystallinity has been increased. The possible reason for this increase in the crystallinity of cellulose by acid treatment could be the hydrogen ions that can enter into the amorphous area of cellulose and destroy the amorphous area. Mechanical grinding and homogenization increased the crystallinity of cellulose due to the non-knotting portion of bagasse bleaching pulp was reduced correspondingly after the high-intensity shearing treatment in the preparation of nanocellulose, and more regular arrangement of fiber in the crystallization area.
Comment 8: As authors have described in morphological discussion, some holes or volumes on fracture surface of films based on load transfer within the films are observed. These defects might have significant impact on mechanical properties during tensile strain. Therefore, authors should add actual plots of mechanical testing (may be provided in Supporting information), if possible, so that clear behavior of the nanocomposite films could be seen.
Response: Thanks for the comments. Some actual pictures of PHB/CNCs have been added and film fracture during mechanical testing to the supporting information. It can be seen from the Figure S3a that the surface of the composite film is flat, but bumps appear in some parts, which is caused by the aggregation of nanocellulose. Figure S3b shows that the composite film was uneven at the fracture during mechanical test fracture, which may be caused by the pull-out of the clusters of CNCs aggregated during fracture.
Figure S3. (a) PHB/CNCs film, (b) Actual picture of PHB / CNCs film fracture during mechanical test.
Comment 9: Authors have described that thermal properties of films were improved with increased CNCs content, but in Figure 14 it is not evident.
Response: Thanks for the suggestions. We have modified the sentence in the revised manuscript and the details can be found in Line 446-447, Page 16. The details are as follows:
“which indicates that the addition of a small amount of CNCs does not significantly improve the thermal performance of PHB”.
In fact, the addition of a small amount of nano cellulose did not significantly improve the thermal performance of PHB (Bulletin of materials science, 40 (2), 383-393, 2017).
Comment 10: In this Figure, PHB/CNCs1 and PHB/CNCs5 show decreased thermal degradation than PHB and PHB/CNCs3. May there be an effect of sulfated-CNCs (as prepared in this study) on properties of films? For this, authors may refer following article: Journal of Materials Physics and Chemistry, 2(1), 2014.
The article (Materials Physics and Chemistry, 2(1), 2014) states that residual sulfate groups cause a reduction in the thermal stability of the nanocrystals. Sulfuric acid molecules were released at much lower temperatures during the degradation process. However, the weight loss curve after adding nano cellulose is basically the same as that of pure PHB at low temperature in Figure 14. Thus, sulfated-CNCs should not affect the thermal performance of PHB.

Reviewer 2 Report
The work by Zhang et al. present an interesting study on the fabrication of biocompatible and biodegradable composites, formed by cellulose and polyhydroxybutyrate. The work presents a big piece of work and is promising for Polymers. However, before publication authors should address several issues:
-Grammar and speling should be revised within the manuscript.
-The quality of the content is not homogeneous, the first part including introduction, abstract and characterization of cellulose presents a poor description. However, the characterization of the composites is well driven and discussed.
-The abstract should summarise in a concise way the aims and main results of the work.
-Authors goes directly from the description of the PHB to the cellulose. However, a short introduction of composites and their interest would increase the understanding of the motivation of this work.
-THe mechanical test in the material and methods section is not clear enough.
-Authors define the aqueous mixtures of their fibers with water as a solution. This is a big mistake. They should use the work dispersion.
-What is the meaning of stratified solution? The image seems to indicate a clear phase separation.
-The good dispersion of nanocellulose in water is not a signature of a size in the colloidal range, what is the meaning of the latter?
-The authors define the distribution obtained from the electronic microscopy images as the size of the nanocellulose in the dispersion. This is not true, What about aggregation upon drying? For this purpose, Dynamic ligth scattering would be useful.
-THe quality of the topographic AFM images is not enough for obtaining conclusion.
Author Response
Date: December 3, 2019
Manuscript ID: polymers-660401
Title: " Effects of Cellulose Nanocrystals and Cellulose Nanofibers on the Structure and Properties of Polyhydroxybutyrate Nanocomposites "
Dear reviewer,
Thank you for your appreciate comments. We find those comments very valuable and helpful, and have taken them into full consideration in our revised manuscript. We have revised the manuscript by adding supporting experiments and related explanations. The main revisions marked in red in the revised manuscript. The point by point responses to the reviewers’ comments are listed below this letter.
With best wishes,
Yours sincerely,
Bobo Zhang, Chongxing Huang and Hui Zhao.
Reviewer #2:
The work by Zhang et al. present an interesting study on the fabrication of biocompatible and biodegradable composites, formed by cellulose and polyhydroxybutyrate. The work presents a big piece of work and is promising for Polymers. However, before publication authors should address several issues:
Comment 1: Grammar and speling should be revised within the manuscript.
Response: Thanks for the kind advice. We have revised the manuscript carefully, and the sentences with grammatical problems have been revised.
Comment 2: The quality of the content is not homogeneous, the first part including introduction, abstract and characterization of cellulose presents a poor description. However, the characterization of the composites is well driven and discussed.
Response: Thanks for the comment. We added some description and characterization of cellulose. The details are as follows:
(1) It has already been utilized for many potential applications such as paper, newspaper, textiles fibers, paper board, construction, etc. [21]. Some researchers have extracted CNCs as nanoreinforcement into bionanocomposite for biomedical and other value-added products in industrial applications [22].
(2) We added a supporting experiment to test the XRD patterns of bagasse cellulose (BC), CNCs and CNFs, as shown in Figure 5.
Figure 5. X-ray diffraction pattern of BC, CNCs and CNFs.
The X-ray diffraction patterns of bagasse cellulose (BC), CNCs and CNFs are shown in Figure 5. When the non-cellulose was removed and the amorphous regions were dissolved, the fibers showed an increased orientation along a specific axis. Typical cellulose I crystal diffraction peaks were 16.4°, 22.5° and 34.5° at diffraction angles 2θ. Three diffraction peaks of the CNCs obtained by the acid treatment and the CNFs obtained by the mechanical method were at the same position, corresponded to the characteristic absorption peaks of the crystal plane of the cellulose I-type crystal structure. This suggested that the acid and mechanical treatments of cellulose did not change the crystal form of cellulose. The diffraction peaks of the two nanocellulose at 22.5° were sharper, and the relative peak intensities were significantly (p≤ 0.05) enhanced, which proves that the crystallinity has been increased. The possible reason for this increase in the crystallinity of cellulose by acid treatment could be the hydrogen ions that can enter into the amorphous area of cellulose and destroy the amorphous area. Mechanical grinding and homogenization increased the crystallinity of cellulose due to the non-knotting portion of bagasse bleaching pulp was reduced correspondingly after the high-intensity shearing treatment in the preparation of nanocellulose, and more regular arrangement of fiber in the crystallization area.
Comment 3: The abstract should summarise in a concise way the aims and main results of the work.
Response: Thanks for the good suggestion. We have rewritten the summary based on the comments. The specific changes are as follows:
One of the major obstacles for polyhydroxybutyrate (PHB), a biodegradable and biocompatible polymer, in commercial applications is its poor elongation at break (~3%). In this study, the effects of nanocellulose contents and their types including cellulose nanocrystals (CNCs) and cellulose nanofibers (CNFs) on the crystallization, thermal and mechanical properties of PHB composites were systematically compared. We explored the toughening mechanisms of PHB by adding CNCs and cellulose CNFs. Results showed that when the morphology of bagasse nanocellulose was rod-like and its content was 1 wt%, the toughening modification of PHB was the best. Compared with pure PHB, the elongation at break and Young's modulus increased by 91.2% and 18.4% respectively. Cellulose nanocrystals worked as a heterogeneous nucleating agent in PHB and hence reduced its crystallinity and consequently improved the toughness of PHB. This simple approach could potentially be explored as a strategy to extend the possible applications of this biopolymer in packaging fields.
Comment 4: Authors goes directly from the description of the PHB to the cellulose. However, a short introduction of composites and their interest would increase the understanding of the motivation of this work.
Response: Thanks for the kind advice. We have added " In addition, as both nanocellulose and PHB are renewable materials, PHB-based biodegradable nanocomposites have great application prospects in the packaging field” in line 40 of the first page.
Comment 5: The mechanical test in the material and methods section is not clear enough.
Response: Thanks for the good comments. The content of mechanical tests in section 2.6.6 has been further improved. The details can be found in Line 149-153, Page 4. As follows:
According to ASTM D882-12, the film (50±8um) was …… equipped with a load cell of 1kN. Five parallel samples were used and the results were averaged.
Comment 6: Authors define the aqueous mixtures of their fibers with water as a solution. This is a big mistake. They should use the work dispersion.
Response: Thanks for the advice. We changed solution to dispersion in the revised article.
Comment 7: What is the meaning of stratified solution? The image seems to indicate a clear phase separation.
Response: Thanks for the advice. Yes, it means phase separation. We have replaced it with phase separation.
Comment 8: The good dispersion of nanocellulose in water is not a signature of a size in the colloidal range, what is the meaning of the latter?
Response: Thanks for the suggestion. Generally, the size of colloidal particles is 1-100nm (according to the diameter of colloidal particles). The diameter of nanocellulose prepared in this study is in this range. We have corrected into colloidal structure.
Comment 9: The authors define the distribution obtained from the electronic microscopy images as the size of the nanocellulose in the dispersion. This is not true, What about aggregation upon drying? For this purpose, Dynamic ligth scattering would be useful.
Response: Thanks for the comments. We added a supporting experiment to test Dynamic light scattering (DLS) measurements in Figure 3.
Figure 3. (a)DLS measurement of the CNCs suspension, (b) DLS measurement of the CNFs suspension, (c) TEM measurement of the CNCs—width, (d) TEM measurement of the CNCs—length, (e) TEM measurement of the CNFs—width.
The DLS result showed a single peak with hydrodynamic diameter of ∼60 nm, indicating the average size of the CNCs (Figure 3a). The CNFs suspension showed a nearly uniform particle size distribution with an average hydrodynamic diameter of ∼106 nm (Figure 3b). Using the particle size analysis Nano Measurer drawing software, 50 nanocellulose from three different TEM images of CNCs and CNFs, were selected and the diameter and length of the nanocellulose were calculated. The average size of the nanocellulose examined by TEM was about 6 or 9 times smaller than that measured by DLS. Similar results were observed while extracting and characterizing nanocellulose from yerba mate sticks and wood pulp [Cellulose, 26(13-14), 2019, 7619-7634; Carbohydrate Polymers, 218, 2019, 78-86.]. This discrepancy could be attributed to the dehydration of the nanocellulose during the sample preparation for TEM [Chemistry of Materials, 28(11), 2016, 4009-4016].
Comment10: The quality of the topographic AFM images is not enough for obtaining conclusion.
Response: Thanks for the comment. We have revised this part as follows:
Figure 4 a–d shows examples of tapping mode AFM images from CNCs and CNFs particles, deposited onto freshly cleaved mica surfaces. Despite a sonication of the suspensions before spreading on mica, some aggregates were observed on the surface. Interestingly, AFM results supported the TEM images of CNCs and CNFs, which showed that the CNCs have a rod-like structure while CNFs have a grid-like structure. Furthermore, the diameter of the two nanofibers prepared were below 100nm.
Round 2
Reviewer 1 Report
In my opinion, this manuscript now can be accepted for publication.
Reviewer 2 Report
the authors have addressed succesfully my main concernings